# Diabetes during Pregnancy: A Maternal Disease Complicating the Course of Pregnancy with Long-Term Deleterious Effects on the Offspring. A Clinical Review

**DOI:** 10.3390/ijms22062965

**Published:** 2021-03-15

**Authors:** Asher Ornoy, Maria Becker, Liza Weinstein-Fudim, Zivanit Ergaz

**Affiliations:** 1Adelson School of Medicine, Ariel University, Ariel 40700, Israel; mariabe@ariel.ac.il; 2Laboratory of Teratology, Department of Medical Neurobiology, Hebrew University Hadassah Medical School, Jerusalem 91120, Israel; liza.weinstein-f@mail.huji.ac.il (L.W.-F.); zivanit@hadassah.org.il (Z.E.); 3Medical Center, Hadassah Hebrew University, Mount Scopus, Jerusalem 91240, Israel

**Keywords:** pregestational diabetes, gestational diabetes, pregnancy, anomalies, growth disturbances, perinatal complications, neurodevelopmental problems, diabetic control

## Abstract

In spite of the huge progress in the treatment of diabetes mellitus, we are still in the situation that both pregestational (PGDM) and gestational diabetes (GDM) impose an additional risk to the embryo, fetus, and course of pregnancy. PGDM may increase the rate of congenital malformations, especially cardiac, nervous system, musculoskeletal system, and limbs. PGDM may interfere with fetal growth, often causing macrosomia, but in the presence of severe maternal complications, especially nephropathy, it may inhibit fetal growth. PGDM may also induce a variety of perinatal complications such as stillbirth and perinatal death, cardiomyopathy, respiratory morbidity, and perinatal asphyxia. GDM that generally develops in the second half of pregnancy induces similar but generally less severe complications. Their severity is higher with earlier onset of GDM and inversely correlated with the degree of glycemic control. Early initiation of GDM might even cause some increase in the rate of congenital malformations. Both PGDM and GDM may cause various motor and behavioral neurodevelopmental problems, including an increased incidence of attention deficit hyperactivity disorder (ADHD) and autism spectrum disorder (ASD). Most complications are reduced in incidence and severity with the improvement in diabetic control. Mechanisms of diabetic-induced damage in pregnancy are related to maternal and fetal hyperglycemia, enhanced oxidative stress, epigenetic changes, and other, less defined, pathogenic mechanisms.

## 1. Introduction

Historically, before the discovery of insulin, a rigorous restricted carbohydrate diet was the main therapeutic strategy for diabetic mothers. At that time, liveborn infants were generally of low birth weight. The purpose of this restricted diet was not only to normalize maternal serum glucose levels but also to prevent or diminish the high rate of low birth weight at term. Maternal starvation was the way to reduce serum glucose levels and avoid intrauterine fetal death or severe diabetic complications with placental damage. Today, everything looks different, but we still have a long way to go for complete satisfaction. 

The prevalence of type 1 or type 2 diabetes mellitus (T1DM or T2DM) in women of childbearing age has steadily increased, affecting about 1% of all pregnancies. The onset of diabetes mellitus in women prior to pregnancy (pregestational diabetes mellitus—PGDM) can produce a variety of adverse effects on the mother, fetus, child, and the course of pregnancy.

Poorly controlled PGDM before conception and during the first trimester of pregnancy is associated with increased rates of major congenital malformations, spontaneous abortions, and increased rate of stillbirth and perinatal mortality [1,2,3,4,5]. PGDM may also be associated with different pregnancy complications and with neurodevelopmental problems in the offspring. Moreover, long-term problems in the offspring resulting from insulin resistance may increase the risk for cardiovascular disease, hypertension, and diabetes (metabolic syndrome).

Gestational diabetes (GDM), which usually develops in the second half of pregnancy, has significantly increased in prevalence over the last 20 years. The current incidence rates are 1.7 to 15.7 percent, depending on the ethnic origin, maternal age, and diagnostic criteria [6,7]. GDM may also cause a variety of pregnancy complications, including increased prenatal and perinatal mortality, perinatal complications, and neurodevelopmental delay [7,8]. 

Of specific note is the possible interference of both PGDM and GDM with fetal growth, often causing increased birth weight (macrosomia). The rate and severity of the above-mentioned complications are in direct relation to the degree of glycemic control. Maximal control will minimize the complications [1,9].

Early diagnosis of PGDM and/or GDM, regular medical follow-up for early identification of complications, stringent glycemic control, and early identification of women at risk for complications is of primary importance. Optimal control will lead to decreased rate of congenital malformations and neurodevelopmental problems, better fetal survival rate, normal birth weight, and minimal negative effects on maternal or fetal health [10,11,12,13,14].

In this review, we will discuss the effects of maternal diabetes in pregnancy (both PGDM and GDM) on embryonic and fetal health, on the health of the newborn infant, and on the long-term neurodevelopment. We will try to emphasize the importance of optimal glycemic control in order to alleviate the different diabetic complications discussed here. However, a full discussion on the mechanisms and pathogenesis of diabetic embryopathy and ways to optimize treatment is beyond the scope of this clinical review. 

## 2. Congenital Malformations and Diabetes in Pregnancy

PGDM is associated with a significantly increased risk for different types of major congenital malformations, with a greater than 10-fold increase in some specific, relatively rare birth defects, compared to nondiabetic pregnancies [15]. Most severe fetal embryopathy is associated with the poorest glycemic control during the first trimester of pregnancy and elevated levels of maternal glycosylated hemoglobin (HbA1c) [16]. 

The most common congenital malformations among offspring of mothers with PGDM include cardiac anomalies, which represent about 40% of the total malformations [17], as well as anomalies of the limbs, neural tube, and musculoskeletal systems [9].

The most frequent cardiac malformations related to PGDM are atrioventricular septal (AVS) defects, hypoplastic left heart syndrome, and persistent truncus arteriosus [15,18,19,20,21]. PGDM is also associated with a higher fetal heart rate during the first trimester [22]. Interestingly, tobacco smoking during pregnancy complicated with PGDM intensifies the diabetic effects on preterm birth and congenital anomalies, especially atrial septal defects, possibly due to the negative effect of tobacco smoking on glycemic control [23].

In a national birth defects prevention study, based on data collected between 1997–2011 in the U.S., Tinker et al. evaluated the association between PGDM and GDM and a range of specific congenital malformations [15]. They detected a statistically significant increased odds ratio for PGDM among 46 out of 50 birth defects evaluated, with point estimates ranging from 2.5 to 80.2. A greater than 10-fold increased risk was observed for sacral agenesis (aOR, 80.2; 95% CI, 46.1–139.3), holoprosencephaly (aOR, 13.1; 95% CI, 7.0–24.5), longitudinal limb deficiency (aOR, 10.1; 95% CI, 6.2–16.5), heterotaxy (aOR, 12.3; 95% CI, 7.3–20.5), persistent truncus arteriosus (aOR, 14.9; 95% CI, 7.6–29.3), atrioventricular septal defect (aOR, 10.5; 95% CI, 6.2–17.9), and single ventricle complex (aOR 14.7; 95% CI, 8.9–24.3). They also observed much weaker associations between GDM and birth defects in 12 out of 56 anomalies evaluated, aOR ranging from 1.3 to 2.1, most of them were cardiac [15].

Several studies have separated between the neonatal outcomes of women with different types of preexisting diabetes. In a population-based study of diabetes during pregnancy in Spain (2009–2015), pregestational type 1 diabetes was associated with more severe neonatal morbidity, preterm birth (RR 3.32; 95% CI 3.14–3.51), and fetal overgrowth (RR 8.05; 95% CI 7.41–8.75) [24].

Wei et al. evaluated the rate of birth defects diagnosed before birth, such as anencephaly, hydrocephalus, open spina bifida, cleft lip, cleft palate, congenital heart disease, and trisomy 21 in a large cohort of Chinese women with impaired fasting glucose (847,737 women) and diabetes (76,297 women) compared to controls (5,523,305 women) [25]. Birth defects were significantly more common among women with PGDM (OR 1.48, 95% CI 1.15–1.91) compared to controls and did not vary between those with impaired fasting glucose and controls (OR 0.95, 95% CI 0.85–1.05) [25].

In a French study based on data of all deliveries taking place in France in 2012, the authors assessed the risk of congenital malformations and adverse perinatal outcomes in GDM [26]. They found an increased risk of cardiac malformations (OR 1.3 [95% CI 1.1, 1.4]) in infants born for women with insulin-treated GDM but not in diet-treated GDM. They also found an unexpected increased risk in fetal mortality in the GDM group (OR 1.3 [95% CI 1.0, 1.6]), which might be due to undiagnosed PGDM [26].

An increased rate of congenital malformations among offspring of GDM mothers was recently also demonstrated in other studies [27,28]. In a retrospective study by Zawiejska et al. [27], the authors analyzed obstetric data of 125 women without PGDM but were considered to have an elevated risk for developing GDM according to Association of the Diabetes and Pregnancy Study Groups (IADPSG) criteria, or presented a fasting glycemia levels above 5.1 mmol/dL during the first trimester. They found that early fasting glycemia diagnostic for GDM (according to the IADPSG criteria) was associated with a significantly increased risk of congenital anomalies, especially cardiac malformations. These pregnant women did not have a diagnosis of hyperglycemia before pregnancy [27].

Wu et al. examined the association of PGDM and GDM with 12 subtypes of congenital anomalies of the newborn among 29,211,974 live births in the U.S [28]. They reported an elevated risk of congenital anomalies at birth with an adjusted relative risk (aRR) of 2.44 (95% CI 2.33–2.55) for PGDM and 1.28 (95% CI 1.24–1.31) for GDM [28]. 

Table 1 describes the rate of increase of some specific malformations in offspring of mothers with PGDM.

Despite the data showing that the primary teratogen in all diabetic pregnancies is maternal hyperglycemia, the underlying mechanisms by which hyperglycemia exerts its teratogenic effects are still not fully understood. Maternal hyperglycemia results in fetal hyperglycemia; the severity of fetal embryopathy depends on the severity and timing of the exposure, genetic predisposition, and metabolic factors. Other proposed mechanisms and causes include involvement of hyperketonemia, elevated metabolism of arachidonic acid, myoinositol, and prostaglandins. Hypoxia and increased oxidative stress have also been reported to be involved in diabetic embryopathy [9,29,30,31]. 

## 3. The Effects of Diabetes on Intrauterine and Postnatal Growth

### 3.1. Growth Disturbances in Newborn Infants of Diabetic Mothers

Fetal growth is a complex multiway process that depends on the maternal substrate supply, uterine environment, and the maternal and fetal hormonal mechanisms that are still to be elucidated. Maternal nutrition, metabolism, maternal illness, and placental function play important roles in normal fetal development and growth that determine the health and disease of the child and adult [8,9,32,33]. 

Many prenatal and postnatal environmental factors (i.e., the severity and onset of diabetes, the degree of diabetes control, and treatment regimens) may affect the growth of the offspring of mothers with diabetes. Postnatal factors that influence the growth of a diabetic mother’s offspring include nutrition, the home environment, health status, and others.

Maternal PGDM and GDM are risk factors for excessive fetal growth (macrosomia) due to maternal hyperglycemia [8,9,32,33,34,35]. The probability of fetal macrosomia is found in about 15–45% of newborns from diabetic mothers and is 3-fold higher than in normoglycemic non-diabetic mothers [36]. It largely depends on the degree of diabetes control. Typically, macrosomia is considered when the birth weight in full-term infants is higher than 4000 gr or, at any gestational age, is higher than the 90th percentile [37]. Fetal macrosomia in diabetic pregnancies is characterized by the larger shoulder and extremity circumferences, a decreased head-to-shoulder ratio, significantly higher body fat, and thicker upper-extremity skin-folds due to the accumulation of subcutaneous fat in the abdominal and interscapular areas [38,39,40]. Macrosomia can firstly be diagnosed by ultrasound around week 24 and is generally maintained throughout pregnancy [40]. Fetal macrosomia has been associated with higher perinatal mortality, risk of Erb’s palsy, shoulder dystocia, brachial plexus trauma, and neonatal morbidity [9,33,41,42,43,44], and is a cause of severe maternal complications during labor that may be threatening maternal life [44,45,46,47]. 

The increased risk of macrosomia in diabetic pregnancy is mainly due to the increased insulin resistance of the mother, which further contributes to maternal hyperglycemia and dyslipidemia [39,48]. High maternal glucose levels increase the trans-placental passage of nutrients to the fetus, resulting in macrosomia [36]. However, the maternal-derived or exogenously administered insulin does not cross the placenta, and the fetus responds to maternal hyperglycemia by hyperinsulinemia, which reduces fetal blood glucose levels, increases fetal adipose tissue, and enhances growth [39]. 

### 3.2. Factors Contributing to Fetal Growth

Generally, pregnancy is a trigger for various maternal adaptation processes that maintain a metabolically healthy embryo ontogeny and growth. Reversible expansion of maternal insulin secretion and progressive insulin resistance is one of the important pregnancy adaptation processes occurring via functional changes and increased β-cell mass (reviewed in references [49,50,51,52]). 

The placenta plays a crucial role in the development of transient insulin resistance in pregnancy that normalizes after birth. Insulin resistance is mediated by the secretion of hormones, cytokines, adipokines, and other substances from the placenta to the maternal circulation [53]. The placenta secretes human chorionic gonadotropin (hCG), human placental lactogen (hPL), and human placental growth hormone (hPGH) that even circumvent the normal hormonal regulatory pathways [54,55]. Placental lactogen and growth hormone increase hepatic gluconeogenesis and lipolysis, and maternal insulin-like growth factor I (IGF-1) levels increase in response to increased growth hormone levels. Placental GH (PGH) is a major regulator of maternal insulin-like growth factor I (IGF-I) [56]. Increased placental levels of IGF-I, IGF-II, IGF-IR, and IGF-IIR mRNA are positively associated with fetal macrosomia [57].

There is an elevation in the levels of pregnancy-associated hormones like estrogen, progesterone, cortisol, and placental lactogen in the maternal circulation [58,59,60], which is accompanied by increased insulin resistance. This usually begins between 20 and 24 weeks of gestation. 

In addition, adipose tissue produces adipocytokines, including leptin, adiponectin, tumor necrosis factor-α (TNF-α), interleukin-6, resistin, visfatin, and apelin [61,62]. These are involved in glucose homeostasis, contributing to insulin resistance in the pregnant woman [62,63]. Pregnant women with GDM have increased glucose levels and small decreases in several amino acids, creatinine, and glycerophosphocholine [64]. Alterations in glucose, amino acids, glutathione, fatty acids, sphingolipids, and bile acid metabolites in mid-gestation in the amniotic fluid of GDM compared to non-GDM fetuses were also described [65]. 

Maternal diabetes is also associated with specific structural placental changes such as increased placental weight, increased angiogenesis (chorangiosis), and delayed villous maturation [66]. The umbilical cord plasma contains glucose, proteins, and lipoproteins that are either released or secreted from the placenta. Some of these substances promote fetal growth and development [67,68]. Hence, fetal macrosomia may be the result of excessive maternal-fetal glucose, amino acids, and fatty acid transfer.

Many substances are thought to be involved in the imbalanced growth of the fetus in diabetes; amongst them are insulin, glucose, leptin, adiponectin, ghrelin, and some growth factors. We will describe the possible associations of these factors to fetal growth and adiposity, as shown in Table 2.

#### 3.2.1. Insulin and Fetal Growth

Insulin, being an anabolic hormone, is involved in the regulation of fetal growth [9,98]. Maternal hyperglycemia induces fetal hyperglycemia and hyperinsulinemia that further stimulates the fetal mitogenic and anabolic pathways in the developing muscles, connective tissues, and adipose tissue [99]. Fetal hyperinsulinemia results in over-growth, whereas fetal insulin-deficiency is associated with intrauterine growth retardation (IUGR) [100,101]. 

High insulin levels were observed in the amniotic fluid of fetuses from mothers with PGDM or GDM [35,72,73]. Carpenter et al. reported on the correlation between second-trimester high amniotic fluid insulin levels in 247 hyperglycemic pregnant women and fetal macrosomia [75]. Hence, maternal glucose intolerance during pregnancy might affect fetal insulin production already in the second trimester of pregnancy. Indeed, it was shown that in the second trimester, the fetuses of diabetic women, in comparison to fetuses of nondiabetics, have pronounced B-cell mass within the pancreatic islets and release more insulin after acute exposure to glucose [102]. The difference in pancreatic B-cell mass between fetuses from diabetic and nondiabetic women becomes more pronounced with increasing gestational age [102]. 

Plasma C-peptide has been found to reflect the insulin-secretory activity of pancreatic β-cells and may be used as a marker of fetal hyperinsulinemia [103]. Fetal hyperinsulinemia has been documented in diabetic pregnancies by analysis of total insulin, C-peptide, and free insulin in umbilical vein plasma. DubÉ et al. found a correlation between cord blood C-peptide levels and maternal levels of insulin, C-peptide, and insulin sensitivity indices values two months after delivery [74]. Cord blood C-peptide levels were measured in a cohort of 18 pregnant women with GDM and in 23 women with normal glucose tolerance (NGT). Higher cord blood glucose levels were noticed in the offspring from GDM mothers compared with those from control nondiabetic mothers. Generally, cord blood C-peptide levels from both groups correlated with maternal insulin, fasting C-peptide, insulin sensitivity, interleukin-6, body mass index, and infants’ weight.

#### 3.2.2. Glucose and Fetal Growth

Fetal macrosomia may be the result of excessive maternal to fetal glucose transfer via aberrant function and expression of glucose transporter proteins (Glut proteins) in the placenta [104]. Increased glucose levels were observed in fetal amniotic fluid samples obtained from GDM mothers [64,65]. In term human placenta from both GDM and PGDM mothers, the increased expression of GLUT-1, GLUT-4, and GLUT-9 correlated with fetal birth weight, demonstrating the role of GLUT proteins in the facilitation of intrauterine fetal growth [105].

A longitudinal ultrasound study of intrauterine growth conducted on 37 fetuses of diabetic mothers and 29 fetuses of non-diabetic, non-smoking mothers demonstrated body compositional changes in hyperglycemic mothers in the second and third trimesters of pregnancy [106]. Fetal macrosomia in mothers with GDM could be managed by an adequate daily intake of carbohydrates (250 g/day) and a reduced fat consumption [107]. Interestingly, in non-diabetic pregnant mothers with glucose levels above 130 mg %, the term infants’ weight was increased by 200 g, while maternal hypoglycemia was associated with reduced birth weight [70]. These studies demonstrated that hyperglycemia increases fetal weight and emphasizes the importance of good glycemic control during pregnancy.

Combs et al. evaluated the effects of fasting and postprandial glucose levels in relation to newborns’ weight in a cohort of 111 women with T1DM and controls from 13 to 36 weeks of gestation [71]. Macrosomia, occurring in 32 infants (29%), was associated with higher postprandial glucose levels at late gestation. Low postprandial glucose values—less than 130 mg/dL—were associated with a higher risk of small for gestational age (SGA) infants (18%). Hence, the authors suggested that optimal 130 mg/dL (7 mM) 1-h postprandial glucose levels should be achieved to avoid risk for fetal macrosomia [71]. Parfitt et al. also defined the association of high postprandial glucose and HbA1 levels to fetal growth and neonatal size in 14 tightly glucose controlled T1DM pregnant women [108].

Li et al. studied the association of GDM and glucose tolerance with fetal growth in a cohort of 107 women with GDM, 118 with abnormal glucose tolerance, and 2020 women with normal glucose tolerance [109]. Glucose levels, even during weeks 10–14 of gestation, were positively associated with larger estimated fetal weight beginning from gestation week 23, reaching a significance at week 27. The authors suggested that measures to reduce fetal overgrowth associated with GDM should begin at weeks 24–28 [109]. This gestational stage is the currently recommended stage for GDM screening in many countries.

Chiefari et al. provided additional evidence for early-onset of fetal overgrowth in correlation to GDM, as they found that fetal bio-metric growth centiles were significantly higher in women with GDM than in women with normal blood glucose tolerance [110]. Moreover, neonates born to mothers with early diagnosis of GDM (during weeks 16–18) demonstrated lower birthweight centiles compared with neonates born to women with late diagnosis of GDM at week 24–28 [110].

Quaresima et al. estimated the significance of timing for GDM screening at 16–18 and 24–28 weeks of gestation to prevent fetal macrosomia in the population-based, retrospective cohort study of 769 pregnant women, stratified into three groups according to the risk factors to develop GDM in the late second trimester [111]. Abdominal circumference and estimated fetal weight in high-risk (HR) women with obesity, history of GDM, or evidence of glucose intolerance during the first trimester of pregnancy were compared to medium-risk (MR) and low-risk (LR) pregnant women and their offspring. Fetal fat deposition and growth rates were significantly higher in MR and LR women with GDM in comparison to normal glucose-tolerant women. However, no significant difference in their infant’s weight percentiles was observed. It was concluded that diagnosis and treatment for GDM at 24–28 weeks of gestation in women with medium and low risk are sufficient to prevent fetal macrosomia [111]. However, infants born to HR women diagnosed with GDM at 24–28 weeks of gestation have higher abdominal circumference estimated fetal weight and higher birthweight compared to normal glucose-tolerant women or MR and LR women with GDM [111]. These studies highlight the importance of early detection of glucose intolerance in pregnant women, even in the first trimester of pregnancy.

Silva et al. studied the association of infants’ birthweight with treatment types used to control glucose levels in 705 pregnancies complicated with GDM [112]. Maternal hyperglycemia was normalized by diet and, in cases when glucose levels remained abnormal, they received metformin and even insulin. Metformin treatment in GDM women decreased the risk for SGA and was associated with appropriate for gestational age (AGA) newborns. Insulin treatment was associated with reduced risk for preterm delivery, whereas metformin treatment combined with insulin was correlated with large for gestational age (LGA) infants [112].

Ladfors et al. investigated the contribution of obesity and gestational weight gain (GWG) to neonatal macrosomia in 221 women with T1DM and in 87 women with T2DM [113]. The occurrence of LGA in women with T1DM was 50%, while in women with T2DM, it comprised only 23%. Gestational weight gain in both T1DM and T2DM mothers strongly correlated with the infant’s overgrowth, while BMI was unrelated. In addition, maternal HbA1c levels were found as a risk factor, particularly in the third trimester, for LGA infants in mothers with T1DM but not in with T2DM [113]. It seems that besides excessive glucose, other substances cross the placenta and contribute to fetal overgrowth [114]. Maternal high amino acid blood levels also correlated with fetal macrosomia [115]. Hence, GWG should be taken with care in diabetic mothers to reduce the risk for fetal overgrowth or other postnatal complications [116,117,118,119,120].

#### 3.2.3. Blood Leptin Levels and Fetal Growth

Leptin, a 167-amino-acid product of the human leptin gene, is considered a satiety hormone secreted primarily by adipocytes [121]. During pregnancy, most of the leptin is produced by the placental trophoblastic cells [122,123]. The exact role of leptin in fetal growth in the environment of maternal diabetes is still unclear. Various studies have contradictory results and different conclusions. Atègbo et al. measured elevated levels of leptin in maternal circulation in gestational diabetic mothers compared with pregnant control women, while leptin levels were lower in macrosomic babies in comparison with their age-matched control newborns [76]. Maternal blood leptin levels correlated with maternal insulin, fasting glucose, and triglycerides. Horosz et al. did not find any difference in maternal plasma leptin levels at later gestation among 86 mothers with GDM compared to 48 controls [79]. Wolf et al. reported that maternal insulin and leptin levels were higher than in fetal circulation. Fetal leptin levels, but not maternal, were associated with LGA infants and fetal insulin [77].

Higher leptin C peptide and insulin levels were found in cord blood of macrosomic infants compared to controls [80,81,124] and were independent of maternal levels [78]. Increased blood cord leptin levels were observed in newborn infants of PGDM mothers, normalizing in the third day of life [82]. Chaoimh et al. found a positive correlation between cord blood leptin at birth and the fat mass index in term infants [83]. However, as findings are contradictory, the infants’ adiposity may not be related to leptin. Indeed, children with congenital leptin gene mutations or leptin receptor gene mutations generally have normal weight at birth [125,126]. It can be concluded that leptin levels are increased in macrosomic fetuses, but the process of interaction with fetal growth is not yet elucidated.

#### 3.2.4. Blood Adiponectin Level and Fetal Growth

Adiponectin, a 30 kDa protein composed of 244 amino acids, is mainly derived from adipocytes, but can also be found in other tissues, particularly in the placenta [127]. Adiponectin is inversely related to leptin blood concentrations and is reduced in obese people [84]. It enhances insulin sensitivity, improves glucose and lipid metabolism, and has anti-inflammatory actions [128]. Generally, the level of adiponectin is reduced in PGDM and GDM in maternal blood during late gestation in comparison to non-GDM women [85] and correlates with high insulin and C-peptide. Lindsay et al. found that fetal cord blood adiponectin levels of PGDM mothers gradually increased with gestational age in comparison to controls [85]. However, other investigators did not find any association between GDM and cord blood adiponectin levels [129,130,131]. It is postulated that fetal adiponectin is expressed, released, and circulates separately from maternal adiponectin circulation. Thus, it is still not clear whether adiponectin plays a significant role in fetal growth.

#### 3.2.5. Blood Levels of Ghrelin and Fetal Growth

Ghrelin, a 28-amino acid peptide, is encoded by the preproghrelin gene Ghrl and primarily secreted by entero-endocrine cells of the gastrointestinal tract and hypothalamus [132,133]. During pregnancy, ghrelin levels in maternal circulation are decreased [134,135], probably as a result of adaptation processes reflected by insulin resistance, and pituitary GH substitution by placental GH [55].

Increased levels of ghrelin expression in term placentae were found in GDM compared to non-diabetic mothers [86,87]. Fetal ghrelin levels were inversely associated with neonatal birthweight [88]. Farquar et al. found that the umbilical cord levels of ghrelin were inversely related to birth weight z- score (adiposity) and to cord blood glucose [89]. They also described a positive correlation between fetal ghrelin plasma levels and gestational age in AGA or LGA infants, but a negative correlation in small for gestational age (SGA) infants [89]. Ng et al. reported reduced fetal ghrelin levels in 38 newborns of mothers with type 1 PGDM treated with insulin, in comparison to 40 infants of control non-diabetic mothers and 42 infants born to mothers with GDM treated only by low energy diet [90]. However, a prospective study by Hehir et al. did not find any significant difference in maternal plasma or fetal umbilical vein plasma ghrelin levels at 36 weeks gestation in a cohort of 10 mothers with PGDM and 10 control [87]. The exact role of ghrelin action in relation to fetal growth is yet unclear.

#### 3.2.6. Human Placental Growth Hormone (PGH)-IGF Axis and Fetal Growth

PGH is secreted by the placental syncytiotrophoblastic cells, appears in the maternal circulation from the sixth week of gestation, and gradually replaces pituitary growth hormone during pregnancy [54]. PGH regulates fetal growth by stimulating gluconeogenesis, lipolysis, and anabolism in maternal tissues, increasing nutrient availability, or via regulation of IGF-I [55,136]. The levels of total PGH and IGF-I in maternal circulation drop in the third trimester in pregnancies complicated by intrauterine growth retardation (IUGR) [91]. Fetal IGF-1 levels are well correlated with neonatal adiposity and may facilitate neonatal fat accumulation [92].

In a prospective study by Higgins et al., maternal PGH was significantly associated with a fetal weight estimated by ultrasonography, birth weight, and birth weight centile in pregnancies with type 1 PGDM [93]. However, PGH levels in maternal and fetal circulations were similar in diabetic and non-diabetic pregnant women, and fetal PGH levels did not correlate with fetal growth [93]. Ringholm et al. found decreased levels of maternal PGH during early pregnancy complicated with type 1 PGDM in women who had large for gestational age infants (LGA), but the levels of maternal PGH were not correlated with insulin in these women [94]. McIntyre et al. suggested that PGH may have a modulatory role in fetal growth via fetoplacental feedback [137].

PGH is considered a regulator of maternal insulin-like growth factor-I (IGF-I) that belongs to the IGF family and facilitates both mitogenic and anabolic pathways [55]. IGFs bioavailability required the association with specific IGF-binding proteins (IGFBPs). Several studies investigated the associations of IGF-I and IGFBPs in maternal plasma in pregnancies complicated with DM. Maternal IGF-I was decreased in diabetic pregnancies [95], whereas fetal IGF1 serum levels were increased in infants of mothers with T1DM and T2DM compared to non-diabetic controls [96]. Higher levels of IGFBP3 were observed in both maternal and fetal serum in T1DM pregnancies compared to non-diabetic controls, probably due to increased proteolysis in maternal but not in fetal serum [93].

McIntyre et al. evaluated growth hormone binding protein (GHBP) levels in maternal circulation in a cohort of 140 women with T1DM, T2DM, or with normal glucose tolerance [91]. A gradual reduction in GHBPs levels in the maternal circulation of non-diabetic mothers was found throughout gestation and was positively correlated with maternal weight and body mass index (BMI) [91]. In contrast, increased levels of GHBPs were detected in the blood of T2DM pregnant women whose infants were SGA. Hence, the decline of GHBPs in maternal circulation may affect fetal growth and maternal glycemic status.

SGA infants have increased levels of IGF binding protein 1 (IGFBP1) in amniotic fluid at the beginning of mid-gestation [138]. Amniotic fluid levels of IGFBP3 are positively associated with birth weight in LGA infants [97].

It can be concluded that fetal growth and development are dependent on various growth factors such as glucose, insulin, PGH and IGF-I, and IGFBP. The IGF1–IGF2 axis may be dysregulated, especially in uncontrolled diabetic pregnancies [106]. The exact role of each of these factors in the interference with fetal growth remains to be elucidated.

Figure 1 is a scheme of the factors influencing fetal growth in diabetic mothers. Insulin resistance is mediated by the secretion of pregnancy-associated hormones like estrogen, progesterone, cortisol, cytokines, and by other growth hormones secreted by the placenta entering the maternal circulation, like an hPGH and placental lactogen. In addition, adipose tissue produces adipocytokines, including leptin, adiponectin, tumor necrosis factor-α (TNF-α), that are possibly contributing to insulin resistance. Generally, insulin resistance induces glucose intolerance, resulting in hyperglycemia. This, in turn, causes placental changes and excessive glucose, amino acid, and lipids in the fetus. The fetus responds to maternal hyperglycemia by hyperinsulinemia, which in turn reduces fetal blood glucose levels but also increases fetal adipose tissue and enhances growth. In addition, PGH facilitates fetal gluconeogenesis and lipogenesis, further contributing to enhanced fetal growth.

### 3.3. Low Birth Weight in Infants of Diabetic Mothers (LBW, SGA)

With the introduction of insulin, the number of low-birth-weight infants (SGA) from PGDM mothers was reduced. Babies with low birth weight generally indicated maternal severe diabetic vascular complications, hypertension, or renal disease [139,140,141,142]. A too rigorous treatment of diabetes may lead to periodic hypoglycemia that may also be a cause of low birth weight [112]. Langer et al. studied the relationship between optimal glycemic control and perinatal outcomes in women with GDM [143]. They found a higher incidence (20%) of SGA infants in women with GDM with mean gestational blood glucose levels below 87 mg/dL in comparison to controls that had only 11% of SGA infants. Like macrosomia, SGA is a risk factor for a variety of diseases, including hypertension, cardiovascular diseases, and diabetes [143,144,145,146,147].

### 3.4. Follow-Up Studies of Weight and Height in Children of Diabetic Mothers

Many perinatal and postnatal factors influence the growth and development of the offspring of diabetic mothers, especially those factors that are primarily associated with nutrition during early life.

Reduced fetal growth has been shown to be associated with an increased risk of insulin resistance, obesity, cardiovascular disease, and type 2 diabetes mellitus [148,149]. Fetal overgrowth (macrosomia) is also associated with significant neonatal and subsequent morbidity. In addition, dysregulation of the GH–IGF axis, adrenal and gonadal function are observed in individuals with abnormal weight gain in infancy and childhood [150]. Barker et al. reported on overweight during childhood and adolescence in children whose weight was low at birth, and they were prone to develop coronary heart disease in later life [151]. GDM is associated with childhood obesity and a distinct growth pattern of LGA rather than of AGA infants born to non-diabetic mothers [152].

#### 3.4.1. Postnatal Growth of Low Birth Weight (SGA) Infants

Generally, small-for-gestational-age (SGA; birthweight < 10th centile) infants demonstrate an accelerated compensatory postnatal growth, a “catch-up growth,” especially in the first year of life, to achieve the genetically predetermined size at adulthood under appropriate nutritional and care conditions [153,154]. This accelerated compensatory growth apparently results from the adaptations of the neuroendocrine system or by more rapid cell proliferation in the skeletal growth plates and in the non-skeletal tissues or both [155,156]. Optimal growth pattern for term SGA infants was suggested as the fast catch-up growth to about the 30th percentile in the first several months, followed by modest catch-up growth around the 50th percentile by 7-years, minimizing the risk for complications at adulthood [157].

There are relatively few studies on the postnatal growth of SGA infants born to diabetic mothers. Boghossian et al. studied the growth of a cohort of 10,781 extremely preterm SGA infants born at 22–28 weeks gestation to diabetic mothers [158]. Women were treated with insulin before pregnancy or started insulin treatment during pregnancy or did not receive insulin. At 18–22 months of age, extremely premature SGA infants of mothers treated with insulin before pregnancy had lower weight, length, and head circumference z score than those of mothers with T2DM that were not treated with insulin [158]. Biesenbach et al. compared the postnatal growth of 10 children of mothers with PGDM complicated by nephropathy to 30 children of mothers with PGDM without nephropathy [139]. Children born to the PGDM mothers with nephropathy demonstrated persistent reduced growth on the follow-up to 3 years of age, and about half of the 10 children had body weight and height below the 50th percentile. The weight and height of children born to PGDM mothers without nephropathy were above the 50th percentile [139].

#### 3.4.2. Postnatal Growth of Appropriate for Gestational Age (AGA) or Macrosomic Infants

Generally, LGA infants tend to be overweight and taller as adolescents and as adults [159,160,161]. In a follow-up study, Silverman et al. found that the higher birth weight observed in over 50% of infants born to mothers with GDM and PGDM normalized up to one year [35]. By age 5, the children showed a significant weight gain, and by age 8, more than 50% of children were overweight at the 90th percentile level or higher with a significantly higher height. Similar results were reported by Rizzo et al. in their studies on children born to diabetic mothers [159] and by Vohr and McGarvey, who also found that LGA newborns of mothers with GDM have increased fatness at one year of age [161]. Tarry-Adkins et al. performed a systematic review and meta-analysis study of infants of GDM mothers, where glucose control was achieved by treatment with metformin versus insulin [162]. Babies whose mothers were treated with metformin weighed on average 108 g less at birth than those whose mothers were treated with insulin. The postnatal follow-up demonstrated that children of metformin-treated mothers were 0.44 kg heavier at 18–24 months and had higher BMI (by 0.8 kg/m^2^) at 5–9 years of age than those of mothers who received insulin [162].

Ornoy and associates studied the weight and height in a cohort of 57 6–12-year-old children born to women with well-controlled PGDM and 32 children of similar age born to mothers with GDM compared to control children of similar ages. Most children had normal birth weight [8,163,164]. The children of diabetic mothers, especially at 9–12 years of age, weighed more and were taller than age and socio-economic status (SES) matched control children. There was no correlation between birth weight and the weight at examination. There was no significant difference between the children of diabetic mothers and the controls in the head circumference.

It can be concluded that children born to diabetic mothers tend to be heavier and taller even if they had normal birth weight. Prevention of macrosomia in diabetes can be achieved by controlling maternal weight prior to pregnancy, weight gain during pregnancy, and precise control of diabetes [165,166].

## 4. The Effects of Diabetes in Pregnancy on the Newborn Infant and in the Neonatal Period

Both PGDM and GDM are associated with significant morbidities in the offspring that are decreased by optimal diabetes control before and during pregnancy [167]. A very recent large retrospective cohort study from the U.S., which was based on data of nearly 200,000 neonates born between 1999–2008, reported that neonates born to mothers with PGDM had 2.27 (95% CI 1.95−2.64) times higher risk for composite severe neonatal morbidity including respiratory distress syndrome and mechanical ventilation, compared to those whose mothers did not have diabetes or to those whose mothers had GDM (aOR 1.96, 95% CI 1.63−2.35). They did not find, however, any association between maternal diabetes and neonatal mortality, while such an association was reported in other studies [167].

### 4.1. Postnatal Complications

#### 4.1.1. Diabetes Associated Stillbirth and Perinatal Death

The association between diabetic pregnancy and stillbirth is well known. Antepartum stillbirth is the death of the fetus before the onset of labor that occurs at or greater than 20-week gestation or at a birth weight greater than or equal to 350 g. Smith et al., in a literature review, found that diabetes was the medical condition most strongly associated with stillbirth [168]. The Odds ratio was 1.7–2.2 for diabetes treated with diet and 1.7–7.7 for insulin-treated diabetes. The increased risk was found among both normal and malformed fetuses [168]. The Scottish Morbidity Record of PGDM found that stillbirth rates were 4.0- and 5.1-fold higher in type 1 and type 2 diabetes compared to the non-diabetic population (*p* < 0.001) [169]. Risk factors for stillbirth in diabetic women included uncontrolled hyperglycemia, obesity, prior cesarean delivery, congenital birth defects, and fetal growth restriction [170]. Syed et al. showed in a meta-analysis of 70 studies that optimal diabetes diagnosis and management yielded a 10% reduction in stillbirths [171]. Reddy et al. compared 712 singleton antepartum stillbirths to 174,097 singleton live births and found that preexisting diabetes had a hazard ratio of 2.7 compared to normal pregnancies [172].

In order to elucidate the pathophysiology of stillbirth in diabetic pregnancies, Bradley et al. obtained fetal blood samples from women with type 1 diabetes between 20–40 weeks gestation by cordocentesis. They found significant acidosis (*p* < 0.001) and hyper-lacticaemia (*p* < 0.01) in the third trimester. Plasma lactate showed significant correlations with PO_2_ but not with birth weight [173].

Perinatal death (defined as a child born after 24 weeks’ gestation who did not breath or show signs of life, or died in the first week of life) was 3.1 times more common among pregnancies with type 1 diabetes and 4.2 times more common among pregnancies with type 2 diabetes in the Scottish Morbidity Record [169]. Perinatal mortality was more common among Chinese women with impaired fasting glucose (847,737 women) and diabetes (76,297 women) compared to controls (5,523,305 women) (OR 1.08; 1.03–1.12; *p* < 0.001) [174]. Chen et al. showed in the Canadian population that the risk elevation in perinatal death for PGDM was much higher in first Nations than non-Indigenous (5.1-fold compared to 1.8-fold). They assumed that the poor glycemic control that was more common among women from the First Nations might partly explain this difference [175].

#### 4.1.2. Time of Delivery

Since poor glycemic control may lead to a greater risk of death during the perinatal period, the timing of delivery is crucial to prevent stillbirth and perinatal death. The recommended gestational age for delivery according to the American College of Obstetricians and Gynecologists (ACOG) depends on diabetes type and degree of control. Well-controlled PGDM: 39 + 0–39 + 6 weeks; PGDM with vascular complications, poor glucose control, or prior stillbirth 36 + 0–38 + 6 weeks; For GDM: well-controlled on diet and exercise 39 + 0–40 + 6 weeks; well controlled on medications 39 + 0–39 + 6 and poorly controlled: late preterm/early term, individualized [176]. Suspected fetal macrosomia is not an indication for induction of labor before 39 + 0 weeks of gestation because there is insufficient evidence that the benefits of reducing shoulder dystocia risk would outweigh the harms of early delivery [43].

To improve pregnancy outcomes and minimize perinatal death, Harper et al. compared perinatal outcomes of planned deliveries at 37, 38, 39, and 40 weeks to expectant management. In 4905 diabetic pregnancies—of them, 1012 insulin-dependent—the risk of perinatal death at any gestational age examined was low (3/1000 births), including patients who were insulin-dependent whose risk was 6/1000 births or fewer [177]. The risk of a composite adverse neonatal outcome that included assisted ventilation >30 min, birth injury, seizures, or 5-min Apgar score ≤ 3 was <2%. They concluded that since diabetic pregnancy is associated with an increased rate of stillbirth, early-term delivery may be a reasonable option for these pregnancies [177].

#### 4.1.3. Mode of Delivery

The mode of delivery in diabetic pregnancy is influenced by the estimated fetal weight (EFW). The ACOG recommends that the prediction of birth weight is estimated by ultrasonography or clinical measurement. Cesarean Section is recommended in diabetic pregnancy with ultrasound (US) EFW over 4500 gr [43].

The common use of US to evaluate fetal weight increased the rate of CS even among pregnancies with lower EFW. Dude et al. found an increased rate of CS among women who had US EFW 5 weeks prior to delivery, even after accounting for birth weight [178]. Among 304 diabetic pregnancies, 231 had US EFW, of them, 66 (28.6%) had the pre-delivery diagnosis of LGA. Following delivery, only 23 (34.9%) of this group were LGA. Among the 165 women with the pre-delivery diagnosis of EFW AGA, 6 (3.6%) were LGA. Pregnancies predicted to have an LGA infant were more likely to undergo a CS with the diagnosis of an arrest disorder [178].

There is no standard for delivery care in diabetic women with retinopathy. Women with PGDM with proliferative retinopathy have an increased risk of vitreous hemorrhage. Ab-delaal et al. assumed that vaginal delivery could be risky in women with severe, non-proliferative, or proliferative diabetic retinopathy due to the Valsalva maneuver and suggested that the C section may minimize the risk of vitreous hemorrhage during vaginal delivery [179].

#### 4.1.4. Shoulder Dystocia

Macrosomia is associated with injuries related to traumatic delivery, especially shoulder dystocia. In a meta-analysis of 41 studies, including 112,034 patients that had third-trimester ultrasounds for the prediction of macrosomia, the EFW did not have a clinically significant effect at predicting shoulder dystocia. Most studies reported sensitivities below 30%, and only one small study reported a sensitivity of >50% [180]. Untreated diabetes in pregnancy was associated with an increased rate of shoulder dystocia (risk ratio, 1.25 compared to controls). Treated diabetes in pregnancy had a shoulder dystocia rate similar to controls even when higher glucose levels were depicted [181]. None of the offspring of 149 women who had pre-pregnancy care suffered from shoulder dystocia compared to 6/265 among those with no pregnancy care (*p* = 0.07) [182]. Kekalainen assumed that pregnancy planning would improve the outcome among women with type 1 diabetes. Women with planned pregnancies had lower HbA1c levels and fewer congenital anomalies; however, there was no difference in the rate of shoulder dystocia, 3/96 in planned pregnancies compared to 3/49 in non-planned pregnancies [183].

#### 4.1.5. Prematurity and Prematurity Complications

The increased rate of prematurity among diabetic pregnancies is associated with multiple pregnancies, increased urinary tract infections, poor glycemic control, preeclampsia, and poor fetal growth. Berger et al. evaluated 30,139 pregnancies complicated by prematurity, of which 7375 had diabetes or diabetes complicated by hypertension or obesity. The relative risk for prematurity was 3.51, 95% CI 3.26–3.78 for diabetes, 6.34, 95% CI 5.14–7.80 for diabetes complicated by hypertension, and 3.09, 95% CI 2.80–3.40 for diabetes complicated by obesity [184]. The highest risk was seen in women with diabetes complicated by both hypertension and obesity—11.26, 95% CI 9.40–13.49 [184]. Riskin et al. evaluated 526 diabetic pregnancies and showed that prematurity was more common in pregnancy with PGDM (31.9%), compared to GDM (11.3%), and among controls only 4.9% (*p* = 0.001) [185].

Soliman et al. also found among 3027 women with GDM and 233 with PGDM that preterm delivery was significantly higher in women with PGDM and GDM (13.7% and 9%, respectively) compared to controls (6.4%); *p* < 0.001) [186]. Antoniou et al. found 8.2% premature births among 576 diabetic pregnancies associated with HbA1c ≥ 5.5% at the end of pregnancy [187]. Prematurity was more common among women with type 1 and 2 PGDM without pre-pregnancy care (17.7% of 265) compared to women with pre-pregnancy care (11.4% of 149), *p* = 0.09 [180]. Kawakita evaluated 222,978 singleton pregnancies, of them, 11,327 (5%) had GDM and 3296 (1.5%) PGDM and found prematurity rate of 16.3% and 32.3% among GDM and PGDM, respectively, compared to controls (10.9%) [188].

#### 4.1.6. Course at the Neonatal Intensive Care Unit (NICU)

Most studies showed that premature infants born to diabetic mothers tend to have a more complicated course in the NICU, mostly due to respiratory morbidity. Battarbee et al. evaluated the outcome of 2993 infants born to mothers with PGDM and 10,549 infants with GDM compared to controls [167]. The median gestational age at delivery was 1-week earlier for neonates born to women with PGDM compared to those with GDM and almost 2 weeks earlier than controls (39.1 weeks) (*p* < 0.001). They showed increased intensive care unit admission (OR 4.89 and 1.68, respectively) due to respiratory morbidity (see below). However, they did not find an association between maternal diabetes and neonatal necrotizing enterocolitis, grade 3 or 4 intraventricular hemorrhage, or death [167].

Hitaka et al. evaluated very low birth weight (VLBW) (<1500 gr) infants in Japan and showed that mortality and morbidity were not significantly different between infants of mothers with or without hyperglycinemia in pregnancy [189]. When 682 infants of mothers with hyperglycemia were compared to 28,944 infants of normoglycemic mothers, there was no statistical difference in the rate of necrotizing enterocolitis (NEC) (*p* = 0.67), patent ductus arteriosus (PDA) (*p* = 0.52), severe intraventricular hemorrhage (IVH), or periventricular leukomalacia (PVL) (*p* = 0.27) and retinopathy of prematurity (ROP) (*p* = 0.27). The incidence of respiratory distress syndrome (RDS) was higher than controls only among infants born in the years 2003–2010, and not those that were born between 2011–2013. In the year 2010, the criteria for hyperglycemia were changed so that the frequency of GDM pregnancies increased twofold to fourfold to 6–12%. The increased risk of RDS observed only in infants of mothers with hyperglycemia in pregnancy diagnosed before the relaxation of the GDM diagnostic criteria reflects the severity of maternal hyperglycemia in that time period [189]. Grandi et al. evaluated the outcome of VLBW infants of diabetic mothers (both GDM and PGDM) from the NEOCOSUR South American Network (Argentina, Brazil, Chile, Paraguay, Peru, and Uruguay) [190]. There were 304 out of 12,146 VLBW pregnancies between the years 2001–2010. After adjustment in the logistic regression analyses, NEC grades 2–3 was the only condition independently associated with the diabetic group of women (OR 1.65). There was no significant difference in the need for mechanical ventilation, PDA, late-onset sepsis, and combined major complications index (death or BPD/IVH grade 3–4/NEC 2–3) (OR 1.08, 1.17, 0.45, 1.01, respectively [190]. Boghossian et al. evaluated the outcome of extremely preterm infants (22 to 28 weeks gestation) of women with insulin use before pregnancy, insulin use only during pregnancy, and controls that were cared for at a Eunice Kennedy Shriver National Institute of Child Health and Human Development Neonatal Research Network center (2006–2011) [158]. Infants of mothers treated with insulin from before pregnancy were at increased risk of NEC (RR 1.55) and of late-onset sepsis (RR1.26) compared to controls. ROP was more common in the pre-pregnancy insulin group than those with insulin only during pregnancy (RR = 1.23). There were no significant differences among the three groups (diabetics and controls) in the risk of PDA, early-onset sepsis (EOS), IVH, PVL, or bronchopulmonary dysplasia (BPD). Infants of mothers with insulin before pregnancy had a smaller average head circumference at term compared to those who started insulin during pregnancy and controls. At 18 to 22 months, there were no significant differences in the individual components of the neurodevelopment index among the three groups [158]. Persson et al. evaluated the morbidity of VLBW infants of diabetic mothers [142]. In seven national networks (Canada, Finland, Israel, Italy, Japan, Sweden, and the United Kingdom) of 76,360 very preterm, singleton infants born between the years 2007–2015, at 24 to 31 weeks gestation with birth weights of less than 1500 g, 3280 (4.3%) of whom were born to diabetic mothers. The risk estimates for RDS, IVH grade 3–4 or cystic PVL, treated PDA, severe ROP, BPD, and NEC were similar between diabetic and non-diabetic pregnancies (OR 1.05, 0.91, 1.01, 1.01, 1.01, 0.94, and 1.1, respectively). The results were consistent among most gestational ages, both sexes, and all national populations assessed [142].

Despite the above data, maternal diabetes was associated with an increased rate of ROP in some studies. Opara et al. showed among neonates weighing < 1500 g that the risk of ROP associated with maternal diabetes was 2.64 (*p* < 0.01) [191]. They assumed that there are common pathogenic mechanisms in ROP and diabetic retinopathy as both are retinal vascular diseases in which there is leakage and/or neovascularization from damaged retinal vessels [191]. Tunay et al. also showed that maternal diabetes increased the incidence of ROP among premature infants > 1500 g and found a 25-fold and 6-fold increase in the risk of ROP and Type 1 ROP regardless of maternal diabetes type [192].

#### 4.1.7. Respiratory Morbidity

Diabetic pregnancy is associated with an increased risk of neonatal respiratory morbidity at all gestational ages. Fetal hyperglycemia and hyperinsulinemia have been suggested as mechanisms for delayed pulmonary maturation. Delayed appearance in the amniotic fluid of phosphatidyl-glycerol, a marker of lung maturity, was associated with poor glycemic control among 621 diabetic pregnancies (261 good glycemic control, 360 poor glycemic control) [193].

Various experimental models were developed to elucidate the link between abnormal glycemic control during pregnancy and the increased risk of respiratory morbidities. McGillick et al. evaluated the effect of intra-fetal glucose infusion on mRNA expression of glucose transporters, insulin-like growth factor signaling, glucocorticoid regulatory genes, and surfactant in the lung of the late-gestation sheep fetuses and showed that despite unchanged number of surfactant positive pneumocytes type B, surfactant protein mRNA expression was reduced in the lung following glucose infusion [194]. Miakotina et al. showed that insulin decreased surfactant protein a gene transcription in human lung epithelial cells [195]. The delay in surfactant system maturation following a significant increase in fetal plasma glucose and insulin concentrations may explain the increased incidence of RDS among diabetic pregnancies.

Kawakita et al. evaluated 222,978 singleton pregnancies, of them, 11,327 had GDM and 3296 PGDM [188]. Neonatal respiratory morbidities were significantly increased among all age groups and were significantly higher among PGDM compared to GDM, and both compared to controls. Since the neonatal respiratory morbidity associated with diabetes was not fully explained by the prematurity-associated physiologic immaturity, they concluded that diabetes itself is a risk factor [188]. Battarbee et al. evaluated neonates born between 24−41 weeks’ gestation from two large multisite U.S. cohorts, the Cesarean Registry, and the Consortium on Safe Labor [167]. When 2993 Infants of PGDM and 10,549 infants of GDM mothers were compared to 196,006 controls, PGDM and GDM were associated with an increased risk for RDS (OR 3.09 and 1.23, respectively) and for mechanical ventilation (OR 2.63 and 1.14, respectively) [167].

#### 4.1.8. Hypertrophic Cardiomyopathy

Myocardial hypertrophy was reported in fetuses and neonates of diabetic mothers. However, its exact incidence in children of diabetic mothers remains unclear since it is asymptomatic in most cases. The fetal heart is affected by hyperinsulinemia and hypoxia. A meta-analysis of 39 studies showed that fetuses of both PGDM and GDM pregnancies had in the third trimester increased intraventricular septal thickness compared to controls [196]. A literature review found an incidence ranging from 13% to 44% among asymptomatic and symptomatic neonates [197]. El-Ganzoury et al. found a correlation between poor maternal glycemic control (HbA1c ≥ 7%) and interventricular septal thickness [198]. To better elucidate the pathogenesis of cardiomyopathy in diabetic pregnancy, Topcuoglu et al. compared 41 infants of diabetic mothers to 51 controls [199]. Echocardiographic and Doppler ultrasound scanning performed in the first three days showed that inter-ventricular septal thickness at diastole, posterior wall thickness (*p* = 0.002), posterior wall thickness in diastole, and left ventricular mass were significantly higher in the IDM group (*p* = 0.02, 0.002, 0.03 and 0.04 respectively). HbA1c, interventricular septum in systole and diastole, and left ventricular mass were in correlation with the oxidative stress indexes [199].

In the last few years, functional echocardiography has been used to evaluate fetuses and neonates subject to hemodynamic changes due to the presence of extra-cardiac conditions. Peixoto et al. evaluated the impact of PGD type 1 (31 pregnancies) and type 2 diabetes (28 pregnancies) on the fetal cardiac function compared to controls (120 pregnancies) by using functional echocardiography and spectral Doppler [200]. Fetuses were evaluated at a gestational age of 30.2, 29.7, and 31.1 weeks. Left ventricular myocardial performance index showed significant association with adverse neonatal outcomes (at least one of: fetal death, neonatal death, Apgar score < 7 at the 5th minute, admission to the intensive care unit, macrosomia, respiratory distress, hyperglobulinemia, hyperbilirubinemia, hypocalcemia, sepsis, and hypoglycemia) (*p* < 0.001) [200].

Patey et al. compared fetal and neonatal cardiac geometry, myocardial deformation, and left ventricular torsion between 21 well-controlled term diabetic pregnancies and 54 normal pregnancies [201]. Of the 21 women with diabetes, seven had insulin-controlled PGDM, and 14 had GDM treated by metformin. Fetuses of diabetic women exhibited significant alterations in cardiac geometry, myocardial deformation, and ventricular function. They had a shorter and narrower left ventricle. Following birth, infants of diabetic mothers had persistent alterations in the left ventricular chamber geometry, with thicker walls and narrower and shorter ventricles and mild tricuspid regurgitation [201]. The cardiac hypertrophy is usually transient, often resolving in two weeks to six months of age; however, the long-term outcome of the fetal and perinatal changes is not completely elucidated. Blais et al. evaluated myocardial relaxation in 3-year-old children in relation to the degree of insulin resistance of their mother during pregnancy [202]. They evaluated 29 children from GDM mothers, 36 children from insulin-resistant mothers (women whose fasting and post-OGTT glucose levels were comprised within the limits of normal), and 41 controls and found that left ventricular mass was normal and comparable between the groups. Compared to the controls, impaired myocardial relaxation was more likely in the GDM and insulin-resistant groups (however, the differences did not reach statistical significance). There were higher median cord blood C-peptide and insulin levels in subjects with impaired myocardial relaxation, but this did not reach statistical significance either [202]. Rijpert et al. found that cardiac dimensions and systolic and diastolic function were normal at 7–8 years of age among 30 offspring of diabetic mothers compared to 30 controls, including three offspring of diabetic mothers who had neonatal cardiac hypertrophy [203]. Neonatal macrosomia and poorer maternal glycemic control during pregnancy were not associated with adverse cardiac outcome [203]. From these studies, it can be concluded that diabetic-induced cardiomyopathy in the newborn is generally transient.

#### 4.1.9. Perinatal Asphyxia

As shown repeatedly, maternal hyperglycemia leads to fetal hyperglycemia and hyper-insulinemia. Data from both experimental and human studies support the hypothesis that fetal hyperglycemia increases the risk of fetal hypoxia. Besides fetal hypoxia, alterations in cardiac structure and function (i.e., hypertrophic cardiomyopathy) and placental abnormalities may further contribute to the risk of neonatal complications among offspring of diabetic mothers. In a systematic review, Huynh et al. showed that placentae from PGDM pregnancies had an increased rate of villous immaturity and increased volume and surface area of parenchymal tissue, while placentae from GDM pregnancies had mostly increased weight [66].

Castelijn et al. evaluated 117 women with type 1 diabetes, 59 women with type 2 diabetes, and 303 women with gestational diabetes, and compared the outcome to 15,260 controls, who delivered between the years 2004–2014 [204]. Women with type 1 and type 2 diabetes were more likely to deliver by unplanned cesarean section compared to controls (OR 3.92 and 3.03, respectively). Mean umbilical artery (UA) pH was lower in the offspring of women with type 1 diabetes. The risk of UA pH < 7.20 or UA pH < 7.10 in type 1 diabetes compared with the control group was significantly increased (OR 1.88 and 3.35 respectively). UA pH < 7.20 and UA pH < 7.10 rates in type 1 diabetes were also increased compared to gestational diabetes, (OR 2.01 and 2.64, respectively). In addition, when fetal distress was the indication of instrumental delivery, higher rates of UA-pH < 7.20 or UA-pH < 7.10 were found in women with type 1 diabetes compared to controls (OR 2.81 and 4.47, respectively). Infants of women with HbA1c levels > 53 mmol/mol were at increased risk of UA pH < 7.20 and UA pH < 7.10 compared to women with HbA1c levels < 42 mmol/mol (OR 2.49 and 3.94, respectively). Excluding prematurity, the risk for NICU admission was higher in type 1 and type 2 diabetes compared to controls [204]. Kawakita et al. showed that diabetic pregnancies were associated with an increased rate of poor neonatal adaptation [188]. There was an increased need for oxygen, bag and mask ventilation, Continuous positive airway pressure (CPAP), intubation, chest compressions, and need for epinephrine injection. Offspring of PGDM had worse outcomes compared to GDM and both, worse than controls, *p* < 0.01 [188]. Cnattingius et al. evaluated the risk of asphyxia in 5941 and 711 infants of type 1 and type 2 diabetic mothers and compared their outcome with 1,337,099 controls also adjusted to maternal obesity [205]. Risks of a low Apgar score (0–6) at 5 min were increased among type 1 and type 2 diabetes compared to controls when adjusted for maternal confounders (OR 2.62 and 1.60, respectively, and 2.67 and 1.25, respectively, when also adjusted for maternal BMI). Rates of severe asphyxia-related neonatal morbidity (defined as either neonatal seizures and/or hypoxic–ischemic encephalopathy) were comparable in the offspring of mothers with type 1 and type 2 diabetes and were markedly higher than in the offspring of mothers without diabetes (OR 3.68 and 4.31, and 3.4 and 2.54 when adjusted to maternal BMI). In all three groups, maternal overweight and obesity were associated with increased risks of a low Apgar score and severe asphyxia-related neonatal morbidity [205].

In order to improve healthcare professionals to monitor patients’ health-related indicators and provide timely medical feedback and guidance to improve the physical and psychological status of patients, website-based systems or mobile terminal devices can be used. In a meta-analysis of 32 randomized controlled trials (RCTs), including 5108 patients with GDM (2581 instructed by telemedicine and 2527 controls), the role of remote communication technologies in improving pregnancy outcome was evaluated [206]. Most studies were conducted in China (21 studies, 65.6%). The meta-analysis showed that telemedicine group had significant improvements in controlling glycated hemoglobin (HbA1c) [mean difference (*p* < 0.01)], fasting blood glucose (*p* < 0.01), and 2-h postprandial blood glucose (*p* = 0.01) compared to controls. The meta-analysis of five of the studies showed the beneficial effect of telemedicine interventions on the incidence of neonatal asphyxia (RR = 0.17, *p* < 0.01) [206].

#### 4.1.10. Neonatal Hypoglycemia

Hypoglycemia is a known complication in infants of mothers with diabetes. Hypoglycemia was more frequent among pregnancies with PGDM compared to GDM and controls (28.9%, 7.1%, and 1.7%, respectively, *p* < 0.001) [185]. Yu et al. performed a meta-analysis of over 40 million pregnancies from 100 studies and found that type 1 diabetes was associated with an increased risk for neonatal hypoglycemia (OR 26.62) [207]. Wahabi et al. found in 880 diabetic pregnancies that pre-pregnancy care may have little or no effect in reducing the risk of neonatal hypoglycemia (RR 0.93) [208]. The use of real-time continuous glucose monitoring throughout pregnancy was evaluated among women with Type 1 diabetes from 31 hospitals in Canada, England, Scotland, Spain, Italy, Ireland, and the USA [209]. Compared to standard capillary glucose monitoring, continuous monitoring resulted in fewer incidences of neonatal hypoglycemia requiring treatment with intravenous dextrose (OR 0.45, *p* = 0.0250) [209]. Antoniou et al. evaluated the offspring of 576 pregnancies with GDM, of them, 10.7% suffered from neonatal hypoglycemia. Maternal treatment requirements were associated with a two-fold higher risk of neonatal hypoglycemia (*p* = 0.032), indicating the severity of maternal disease [187].

#### 4.1.11. Fetal Macrosomia (LGA)

Fetal hyperinsulinemia is the main cause of fetal overgrowth in diabetic pregnancies. Among 280 pregnancies in Sweden, 53% of the children of women with PGDM were LGA compared to only 13% in the GDM group (*p* = 0.0001) [210]. Women with type 1 PGDM had a 9.5 times higher risk for having LGA children compared to women with GDM. In multiple regression analyses, the risk for macrosomia was significantly increased among women with diabetes type 1 (OR 31.3, *p* < 0.001); multiparity (OR 6.2, *p* = 0.003): early pregnancy BMI > 30 kg/m^2^ (OR 7.2, *p* = 0.003); gestational weight gain ≥ 8 kg (OR 3.8, *p* = 0.047) and living alone (OR 18.4, *p* = 0.02) [210]. Wahabi et al., found in a meta-analysis that pre-pregnancy care of diabetic women had little or no effect in reducing macrosomia rate (RR 1.06; nine studies, 2787 women) [208]. The different definitions of GDM (i.e., the Diabetes and Pregnancy Study Group (IADPSG) criteria versus the National Institute for Health and Care Excellence (NICE) in the U.K criteria) affect the rate of macrosomia. Koivunen et al. evaluated the outcome of 4033 women screened for GDM [211]. The proportion of GDM was 2.4-fold higher when diagnosed by the IADPSG (31.0%) criteria as compared to that diagnosed by the NICE criteria (13.1%). The rate of either Large for gestational age > 90% and Large for gestational age, > + 2 SD did not differ between treated diabetic pregnancies and controls, irrespective of which diagnostic criteria were applied. Mild untreated hyperglycemia evaluated by both criteria was associated with higher birth weights (OR LGA >90% IADPSG 1.51 and NICE 1.43) and LGA > + 2 SD (IADPSG 1.17 and NICE 1.18) [211].

#### 4.1.12. Fetal Growth Restriction (FGR)

Pregnant women with diabetes mellitus have an increased risk of hypertensive disorder, which affects fetal growth. In pregnant women with DM and hypertension, fetal growth might be determined by the balance between the increased blood glucose supply from the mother to their fetus via a heavier placenta and fetal hyperglycemia and hyperinsulinism, and the decreased blood glucose supply due to placental dysfunction in women with hypertension. Among 6,447,339 pregnant Chinese women, there were 14.23% (924,034) women with pre-pregnancy fasting blood glucose abnormalities: of them, 1.18% (76,297) had PGD, and 13.15% (847,737) had impaired fasting glucose (5.6–6.9 mmol/L). Compared with women with normal blood glucose, women with impaired fasting glucose and diabetes had increased risk for SGA infants (OR 1.06 *p* = 0.007, and 1.17 *p* = 0.008, respectively). Linear associations were observed between fasting blood glucose and SGA (*p* = 0.001). Morikawa et al. evaluated 7893 women, of whom 154 had PGDM (type 1 DM 45 and Type 2 DM 109 women) [212]. Birthweights were significantly higher among women with type 1 DM compared to type 2 DM (*p* < 0.05); however, the frequency of FGR was similar between both diabetic groups. Women with type 1 diabetes had a similar rate of FGR regardless of hypertension in pregnancy, but women with type 2 diabetes had a significantly higher rate of FGR when the pregnancy was complicated by hypertension (*p* < 0.05) [212]. Among 576 women with GDM, 9.4% of the newborns were SGA. Pre-pregnancy BMI showed an inverse association with SGA [187]. In a study of 231 women with GDM, the incidence of newborns with malnutrition (ponderal index (PI) < 10th centile) was 8.7%. The risk of presenting a PI < 10th centile in GDM newborns classified as SGA by customized curves was 4.24 times higher than that of newborns classified as AGA (RR 4.24) [213]. Wahabi et al. found in a meta-analysis that pre-pregnancy care of diabetic women resulted in a large reduction in SGA (RR 0.52; six studies, 2261 women) [208]. Pre-pregnancy care was associated with improved glycemic control in the first trimester of pregnancy. They hypothesized that the reduction in SGA infants was due to a reduction in congenital malformations and healthy lifestyle promotion, including smoking cessation, weight control, and avoidance of teratogenic drugs [208].

#### 4.1.13. Polycythemia

Polycythemia and increased nucleated red cells in cord blood occur mainly in PGDM and not in GDM. Fetal erythropoietin levels are frequently elevated in type 1 diabetic pregnancies to adapt to chronic hypoxia by increasing the oxygen-carrying capacity in the blood. The amniotic fluid erythropoietin concentration in 156 type 1 diabetic singleton pregnancies at a median time of 1 day was independently related to low umbilical artery pH (<7.21; *p* < 0.0001), neonatal hypoglycemia (*p* = 0.002), umbilical artery pO2 (<15.0 mm Hg) (*p* < 0.0001), and fetal macrosomia and growth restriction (*p* = 0.004) [214].

Table 3 summarizes the different perinatal complications in infants of diabetic mothers.

## 5. Development of Children Born to Mothers with Diabetes

Many experimental in vivo and in vitro studies have shown that enhanced embryonic and fetal oxidative stress is a major mechanism of diabetes-induced embryotoxicity and teratogenicity [29,32,215]. Increased oxidative stress in the brain was also described by many investigators to be present in various neurobehavioral and psychiatric disorders. Hence, it is not surprising that PGDM is associated with a variety of neurodevelopmental problems in childhood. Moreover, since the major developmental events of the cerebral cortex occur in utero during the third trimester of pregnancy, as it is characterized by intensive synaptogenesis, dendritic arborization, and neuronal determination [33,216,217], it is expected that gestational diabetes will have similar effects. Indeed, several neurodevelopmental problems related to maternal diabetes were reported in infants of mothers with PGDM as well as of mothers with GDM. Moreover, many investigators examined concomitantly the postnatal development of children born to both types of diabetic mothers [8].

### 5.1. Development of Children Born to Mothers with GDM

Among the first investigators evaluating the possible neurodevelopmental outcome of children born to mothers with diabetes in pregnancy were Persson and Gentz [218] and Rizzo et al. [159], who found no differences in cognitive measures in children born to mothers with PGDM or GDM.

Later, Ornoy et al. studied the gross and fine motor abilities in a group of 32 children, 6–12 years old, who studied in regular schools, born to mothers with GDM, and 57 children born to mothers with PGDM compared to 57 control children [163,164]. The children born to diabetic mothers had significantly lower scores on the Bruininks–Oseretsky fine and gross motor scores as compared to controls. A negative correlation was found between the percent of HbA1c, maternal acetonuria, and the total motor scores, and the sensory-motor function of children born to diabetic mothers was lower with higher glycosylated hemoglobin levels. In addition, there was a higher rate of inattention and attention deficit hyperactivity disorder (ADHD) and a slightly lower verbal intelligence quotient (IQ) [8,164].

Torres-Espinola studied the neurodevelopmental outcome at 6 and 18 months of 79 children born to mothers with GDM compared to 132 control children [217]. While at 6 months the differences from controls were minimal—a tendency towards slightly elevated scores on language—at 18 months, there was a clear tendency towards reduced scores in motor development.

The fact that slight brain damage may occur in children born to mothers with GDM has important clinical implications since GDM is quite common and poorly treated GDM may cause significant metabolic dysfunction, leading to a possible increase in the rate of slight developmental disorders in the offspring.

Ornoy et al. found that serum from women with GDM is teratogenic to 10.5-day old rat embryos in culture to the same extent as serum from women with PGDM, causing over 40% of embryonic anomalies [219]. This implies that the reason for the lack of a significant increase in the rate of congenital anomalies in GDM is the fact that it develops after major organogenesis had occurred. Indeed, early initiation of GDM may also increase the rate of congenital malformations [27,28]. Moreover, since the brain develops actively throughout pregnancy, cerebral hemisphere functions may be affected by the GDM-induced metabolic disturbances even in the second half of pregnancy [30].

### 5.2. Development of Children Born to Mothers with PGDM

Generally, the results on IQ tests of children born to mothers with PGDM are related to the degree of diabetic control, and poor glycemic control may be associated with a slight decrease in the intelligence of the offspring. This might be the reason for the reports on decreased cognitive function in children of mothers with PGDM alongside studies that found normal cognitive function.

#### 5.2.1. Studies Describing Decreased Cognitive Function in Children of Mothers with PGDM

Churchill et al. were among the first to report on lower IQ scores in children born to mothers with PGDM and acetonuria, as opposed to children of diabetic mothers without acetonuria that functioned normally [220]. There was no correlation of the IQ with the duration of maternal diabetes. Stehbens et al. found that children born to diabetic mothers and who were small for gestational age had lower cognitive scores compared with controls [221]. Similarly, Petersen et al. found that SGA children of diabetic mothers had a lower verbal performance at 5 years of age [222]. Bloch-Petersen also found that children born to diabetic mothers with low birth weight and prematurity have at 4–5 years of age an increased risk of poor developmental performance in language, speech, and motor development as assessed by the Denver Developmental Screening test [223]. Kimmerle et al. studied the development of 36 children born to mothers with PGDM and nephropathy (White’s class F), with 10 of the mothers having renal failure [224]. Seven of the children (19%) had moderate to severe developmental retardation and one (3%) had severe motor impairment [224]. Hod et al. studied the development of 31 one-year-old infants, born to mothers with type 1 or type 2 PGDM and found decreased psychomotor abilities in comparison to 41 infants of non-diabetic women [225]. Nelson et al. observed abnormal hippocampal-based recognition memory in infants of diabetic mothers [226]. However, their neurodevelopmental scores on the Bayley scales were not different from that of control infants. An important confounder could be the SGA, as infants with SGA without maternal diabetes also exhibit a variety of neurodevelopmental problems [227].

Camprubi Robles et al. carried out a meta-analysis of 12–15 studies that reported the neurodevelopment of infants and children from 1–14 years of age [228], combining the data for children born to mothers with PGDM and those born to mothers with GDM. There were 6140 children of different age groups starting from one year to 14 years of age. They found a significant decrease in mental scores in infants born to diabetic mothers at 1–2 years of age. Similarly, they reported a reduction in IQ of school-age children born to diabetic mothers, but the data were inconclusive due to large heterogeneity [228]. It is unknown how many of these children were SGA. See Table 4 for additional data.

#### 5.2.2. Studies That Did Not Find Decreased Cognitive Function in Children of Mothers with PGDM

Many neurodevelopmental studies conducted on children born to well-treated mothers with PGDM did not demonstrate any cognitive impairment. Cummins and Norrish [229], as well as Persson and Gentz [218], did not find differences in cognitive scores of children born to diabetic mothers at preschool and early school age. Rizzo et al. did not find a developmental delay in children born to mothers with PGDM or GDM, but found at 6 to 9 years of age a significant negative correlation between maternal second and third-trimester hydroxybutyrate blood levels and motor development, as the children of mothers with high hydroxybutyrate blood levels had lower scores on the Bruininks–Oseretzky test that measures fine and gross motor abilities [159]. Similarly, Sells et al. found normal development of 70 infants at 6–36 months of age born to mothers with type 1 PGDM who started early treatment [230]. However, 39 infants born to PGDM mothers who started treatment late in pregnancy, with poor diabetic control and high glycosylated hemoglobin levels, scored less well than controls on language measures. Ornoy et al. found normal cognitive function on WISC—R in a group of 57 early school-age children of well-controlled mothers with PGDM compared to 57 age-matched children born to non-diabetic mothers [8,163].

**Table 4 ijms-22-02965-t004:** Neurodevelopmental problems in children born to mothers with PGDM.

Type of Neurodevelopmental Problem	Authors and Reference Number	Description of Findings
Decreased cognitive abilities	Churchill et al. [220]	Reduced IQ scores in children of PGDM with acetonuria but not without acetonuria. No correlation with duration of diabetes
Decreased cognitive abilities and language development	Stehbens et al. [221] Petersen et al. [222] Bloch-Petersen et al. [223]	A decrease in children that were SGA. Children’s age ranged from 1–5 years
General developmental retardation including motor delay	Kimmerle et al. [224]	Mothers had PGDM with nephropathy. Most severe problems in offspring of mothers with renal failure
Neurodevelopmental delay in the first year	Hod et al. [225]	Neurodevelopmental delay at one year of age
Normal cognitive development	Persson and Gentz [218] Cummins and Norish [229] Boghossian et al. [158] Rizzo et al. [159] Ornoy et al. [163]	All PGDM mothers were well treated. Infants, preschool and early school-age children. If treatment started late in pregnancy, there were some children with developmental delays
Motor developmental delay	Ornoy et al. [8,163] Rizzo et al. [159]	Children at early school age of well-treated mothers. They had normal cognition
ADHD	Nomura et al. [231] Ornoy et al. [8] Xiang et al. [232] Kong et al. [233] Li et al. [234]	Mothers with PGDM, well treated. Low SES increased the rate of ADHD. More severe PGDM, more ADHD. Obesity with PGDM further increased the rate of ADHD
ASD	Kong et al. [233] Li et al. [234] Xu et al.; meta-analysis of 12 studies [235]	The rate of ASD was further increased if the PGDM mothers were also obese. A similar increase in ASD was also found in offspring of mothers with GDM [234,236,237]

Fraser et al. studied the cognitive abilities of male youngsters at 18 years of age born to women with PGDM or GDM using the large Swedish national birth registry of years 1988–1998 and 1998–2009 [238]. They found that among non-siblings, maternal PGDM and GDM slightly but significantly reduced the intelligence quotient by 1.36 points, but no significant difference was found among sibling discordant for maternal diabetes. The authors concluded that the slight reduction of IQ found in the non-sibling youngsters born to diabetic mothers does not result from maternal diabetes because no such difference was found among the siblings. The authors also state that the negative association among siblings may result from the existence of undiagnosed maternal diabetes among the “unexposed siblings,” and if those are added to the exposed group, the difference might be significant [238].

Boghossian et al. assessed the neurodevelopmental outcome of 536 extremely premature infants (weeks 22–28 of gestation) born to mothers with insulin-dependent diabetes mellitus at 18–22 months corrected age and found no difference in their neurodevelopmental outcome when compared to extremely premature infants born to non-insulin diabetic mothers [158].

It can be concluded that maternal GDM and PGDM do not seem to affect cognitive function in the offspring unless there are diabetic complications that also induce significant fetal growth restriction (Table 4).

## 6. Diabetes in Pregnancy and ADHD

Nomura et al. compared the development of 21 children born to mothers with GDM to 191 children born to non-diabetic mothers and found a two-fold risk for ADHD in the children born to mothers with GDM [231]. Low socio-economic status (SES) further increased that risk and children exposed to GDM and low SES also had lower IQ and poorer language abilities, in addition to an increased rate of ADHD [231].

Ornoy et al. found a high rate of ADHD among 57 children born to mothers with PGDM and 32 children of mothers with GDM [8,163,164]. They found a higher number of failure sores on the Pollack Taper test that measures the attention span and learning ability, as well as on the Conner’s abbreviated Parent Questionnaire in the children born to diabetic mothers compared to controls. The rate of ADHD correlated with the levels of maternal HbA1c.

Xiang et al. assessed the risk for ADHD of both PGDM and GDM [232]. They studied the rate of ADHD in a large group of children (8344 offspring of mothers with PGDM and 29,534 mothers with GDM) and generally found no significant increase. However, when they assessed the rate of ADHD only in children born to mothers who were treated during pregnancy because they apparently had more severe PGDM or GDM, they found a significant increase in the rate of ADHD. Hazard Ratio (HR) was 1.57 (for type 1 PGDM) or 1.43 (for type 2 PGDM) and 1.26 for children of mothers with GDM. HR for children of women with GDM not requiring treatment was 0.93 [232]. These data are similar to our data [8,163,164] as they show the correlation with the severity of PGDM or GDM.

Kong et al., using the live birth registry in Finland, assessed the risk for ADHD and ASD (as well as other psychiatric disorders) among the children of a large number of women with PGDM (4000) and GDM (101,696) with and without obesity [233]. Maternal PGDM and GDM without obesity only slightly increased the rate of ADHD among the offspring (PGDM—HR 1.45; 95% CI 0/98–2.19; GDM; HR 1.15, (95% CI 1.01–1.30) [238]. However, maternal obesity and PGDM further increased the risk of ADHD and conduct disorder (HR = 6.03). It can be concluded that ADHD is increased among the offspring of mothers with PGDM and GDM and the increase is more prominent in diabetes with complications (Table 4).

## 7. Diabetes in Pregnancy and ASD

Of the many studies on the neurodevelopmental outcome of children born to mothers with PGDM or GDM, a significant number assessed different neuropsychiatric problems, including Autism Spectrum Disorder (ASD). Indeed, most studies demonstrated a positive association between GDM, PGDM, and increased rate of ASD in the offspring [239].

Lyall et al. assessed the possible association of maternal GDM and ASD in 793 children with ASD from a cohort of 66,445 pregnancies and found an OR of 1.76 [236]. Gardener et al. found while assessing possible associations of a variety of antenatal maternal factors with ASD, that maternal diabetes was among the leading factors associated with ASD, with an odds ratio of 2.07 (95% CI 1.24–3.47) [237]. Several other studies have shown a similar association. Nahum Sacks et al. studied the neuropsychiatric morbidity in offspring of 12,642 women with gestational diabetes compared to 218,629 non-diabetic women the investigators found an adjusted odds ratio of 4.4 (95% confidence interval: 1.55–12.69) for ASD [240]. Li et al., using the Boston Birth Cohort, found that both maternal PGDM and obesity were highly associated with ASD in the offspring. The hazard ratio (HR) for ASD was 3.91 (95% CI—1.76–8.68), and in obesity and GDM, the HR was 3.04 (95% CI 1.21–7.63) [234]. Kong et al., in their study using the birth registry in Finland, found an HR of 3.64 for maternal PGDM and obesity and an HR of 1.56 for maternal GDM and obesity [233].

There are also several studies that found no associations of diabetes in pregnancy and ASD. For example, Hultman et al., in their case–control study on 408 Swedish children with ASD compared to 2040 matched controls, found an association of ASD with a variety of pregnancy-associated factors but not with maternal diabetes [241]. Xiang et al. found in a study on pregnant women with preexisting type 2 diabetes an insignificant correlation between PGDM and ASD with an OR of 1.21 [242]. However, in the offspring of women with GDM diagnosed before the 26th week of pregnancy and generally more severe, the HR was significant—1.42 (95% CI 1.15–1.74). Kong et al. found no association in non-obese mothers with PGDM or GDM and ASD in their offspring. The additional presence of obesity significantly increased the rate of ASD [233].

Reviews and meta-analyses: Guinchat et al. did not find a strong association of maternal diabetes and ASD in their review of 85 studies [243]. In contrast to this review, Xu and associates culled twelve studies [235]. For the 3 cohort studies, the pooled risk of maternal diabetes (GDM and PGDM) was 1.48 (1.25–1.75, *p* < 0.001), and for the 9 case-control studies, it OR was 1.72 (1.24–2.41, *p* = 0.001). The OR for offspring of mothers with GDM, although significant, was generally lower than that of mothers with PGDM [235].

The mechanism of the association between diabetes and ASD is largely unknown. We should remember that the increased risk found may be related to a variety of pregnancy complications that are common in diabetes, and an association of pregnancy complications to ASD in the offspring is known [9,239]. Judging from the proposed mechanisms of the effects of maternal diabetes on the embryo and fetus, the increase in the rate of ASD in diabetic pregnancies may result from increased fetal oxidative stress, from epigenetic changes in the expression of several genes, and it may also be related to the other neuro-developmental changes induced by maternal diabetes [9]. Optimal glycemic control is probably also the best way to reduce diabetic-related ASD [239,242]. There seem to be no studies relating the prevalence of ASD to the degree of diabetic control.

Table 4 summarizes the different neurodevelopmental problems in infants of diabetic mothers.

## 8. Conclusions

It may not be surprising that similar to several other maternal chronic diseases in pregnancy, diabetes may have a significant negative impact on the developing embryo and fetus and on the course of pregnancy. PGDM is considered to be more disadvantageous for the developing embryo and fetus compared to GDM, but poorly controlled GDM, especially if there are other risk factors such as obesity, might also negatively affect the fetus, inducing similar perinatal complications: changes in growth pattern, neurodevelopmental deviations and, apparently, as a result of early glucose intolerance, also increased rate of congenital malformations. Hyperglycemia is apparently the main etiologic factor behind all these complications, as stringent glycemic control alleviates many of these complications. However, even optimal control of diabetes does not prevent all possible complications. Prevention of diabetes seems to be the best therapeutic measure. Unfortunately, as a result of non-optimal lifestyle, we witnessed in the last twenty years a constant increase in the rate of diabetes with a gradual decrease in the age of onset. It seems that we know quite well how to extinguish the fire, while it is much more difficult to prevent it.

## Figures and Tables

**Figure 1 ijms-22-02965-f001:**
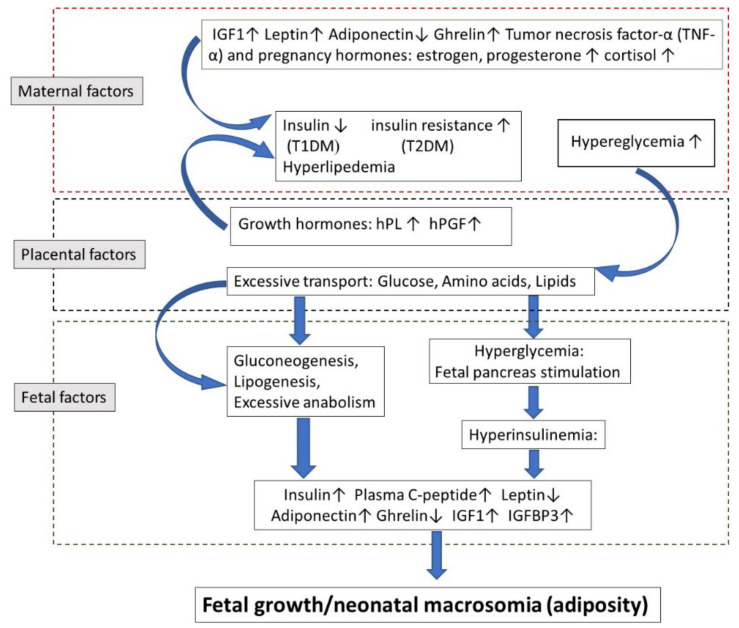
Factors affecting fetal growth in diabetic pregnancies. Insulin resistance is mediated by the secretion of pregnancy-associated hormones like estrogen, progesterone, cortisol, cytokines, and by other growth hormones secreted by the placenta entering the maternal circulation, like hPGH and placental lactogen. In addition, adipose tissue produces adipocytokines, including leptin, adiponectin, tumor necrosis factor-α (TNF-α), that are possibly contributing to insulin resistance. Generally, insulin resistance induces glucose intolerance, resulting in hyperglycemia. This, in turn, causes placental changes and excessive glucose, amino acid, and lipids in the fetus. The fetus responds to maternal hyperglycemia by hyperinsulinemia, which in turn reduces fetal blood glucose levels but also increases fetal adipose tissue and enhances growth. In addition, PGH facilitates fetal gluconeogenesis and lipogenesis, further contributing to enhanced fetal growth.

**Table 1 ijms-22-02965-t001:** Types of congenital malformations in the offspring of mothers with pregestational diabetes mellitus (PGDM).

Type of Birth Defect	Authors and Reference Number	Description of Findings
Cardiac anomalies	Tinker et al. [15]	24 types of cardiac defects showed a statistically significant increased risk for infants born to mothers with PGD, 4 of them with OR greater than 10: truncus arteriosus, heterotaxy, atrioventricular septal defect, and single ventricle complex.
	Wren et al. [19]	transposition of the great arteries, tricuspid atresia, and truncus arteriosus is greater than threefold excess
	Martínez-Frías et al. [18] Corrigan et al. [20] Correa et al. [21]	
Limb anomalies	Tinker et al. [15]	A greater than 10-fold increased risk was observed for longitudinal limb deficiency (aOR 10.1)
	Correa et al. [21]	longitudinal limb deficiencies (aOR 6.47)
musculoskeletal systems anomalies	Tinker et al. [15]	Out of 46 birth defects associated with PGDM, the largest odds ratio was observed for sacral agenesis (aOR, 80.2)
Central nervous system anomalies	Tinker et al. [15]	A greater than 10-fold increased risk for Holoprosencephaly (aOR, 13.1)
	Correa et al. [21]	PGD was associated with anencephaly and craniorachischisis (aOR, 3.39); hydrocephaly (aOR 8.80), anotia/microtia (aOR3.75)
Other birth defects	Correa et al. [21]	Cleft lip with or without cleft palate (aOR, 2.92), anorectal atresia (aOR, 4.70), bilateral renal agenesis/hypoplasia (aOR, 11.91)

**Table 2 ijms-22-02965-t002:** Factors contributing to fetal macrosomia in diabetic pregnancy.

Factor	Maternal Circulation and Placenta	Fetal Circulation (Umbilical Vein or Amniotic Fluid)
Glucose	High glucose levels [64,65]. Increased expression of GLUT-1, GLUT-4, and GLUT-9 in term placenta [69,70,71].	High glucose levels
Insulin	High insulin levels [35,72,73]. Elevated plasma C-peptide [74]	High insulin levels [35,72,73,74,75]. Elevated cord blood C-peptide [74]
Leptin	Elevated levels of leptin [76,77,78]. No differences in diabetic and non-diabetic pregnant women [79].	low leptin levels in macrosomic infants [76,77,78,80,81,82,83].
Adiponectin	Reduced adiponectin levels [84].	Increased adiponectin levels in cord [85].
Ghrelin	Increased levels of ghrelin expression in term placenta [86,87]. No difference in ghrelin levels in diabetic and non-diabetic pregnant women [87].	Decreases ghrelin in cord [88,89,90]. No differences in ghrelin levels in cord blood of neonates from diabetic and non-diabetic pregnant women [87].
PGH-IGF axis	Low levels of total PGH and IGF-I are correlated with IUGR [91]. High IGF-1 levels correlate well with fetal adiposity [92]. High PGH levels correlate well with LGA neonates [93]. No differences in PGH levels in diabetic and non-diabetic pregnant women [93]. Decreased levels of PGH during early pregnancy correlated with LGA infants [94].	No differences in PGH levels in the newborn of diabetic and non-diabetic pregnant women [93].
IGFs and IGFBPs	Decreased levels of IGF-I [95,96]. Higher levels of IGFBP3 [93]. Increased levels of IGFBPs correlated with SGA infants [91].	Increased levels of IGF-I [95,96]. Higher levels of IGFBP3 [93]. Amniotic fluid levels of IGFBP3 well associated with LGA infants [97].

**Table 3 ijms-22-02965-t003:** The effects of diabetes in pregnancy on the newborn infant.

Perinatal Complications	Authors and Reference Number	Description of Findings
Asphyxia	Castelijn et al. [204] Cnattingius [205] Xie et al. [206]	Women with type 1, 2, and gestational diabetes had an increased rate of cesarean section, instrumental delivery, low Apgar score, seizures, and/or hypoxic–ischemic encephalopathy, in association with poor glycemic control.
Macrosomia	Wahabi et al. [208] Koivunen et al. [211] Stogianni et al. [210]	Macrosomia rate increased among women with diabetes, multiparty, obesity, and elevated gestational weight gain. Macrosomia rate decreased with improved glycemic control but was not affected by pre-pregnancy care.
Fetal growth restriction	Wei et al. [25] Antoniou et al. [187] Wahabi et al. [208] Morikawa et al. [212] Fernández-Alba et al. [213]	The frequency of fetal growth restriction was similar between type 1 and 2 PGDM and increased with impaired fasting glucose and obesity. Hypertension increased fetal growth restriction rate in women with type 2 diabetes. Pre-pregnancy care associated with improved glycemic control reduced the fetal growth restriction rate.
Shoulder Dystocia	Moraitis et al. [180] Shah et al. [181] Egan et al. [182] Kekäläinen et al. [183]	Ultrasonographic evaluation of expected fetal weight did not significantly predict shoulder dystocia. Untreated diabetes in pregnancy was associated with an increased rate of shoulder dystocia. Well-treated diabetes in pregnancy had a shoulder dystocia rate similar to controls.
Prematurity	Persson et al. [142] Boghossian et al. [158] Battarbee et al. [167] Hitaka et al. [189] Grandi et al. [190] Opara et al. [191] Tunay et al. [192]	Diabetes in pregnancy increases the prematurity rate. The association between maternal diabetes and morbidities associated with prematurity vary between studies: no association [142,167], respiratory distress syndrome [189], necrotizing enterocolitis [158,190], late-onset sepsis [158], retinopathy of prematurity [158,191,192].
Respiratory morbidity	Battarbee et al. [167] Kawakita et al. [188]	Respiratory morbidities were significantly increased among all age groups and were significantly higher among PGDM compared to GDM, and both compared to controls.
Hypertrophic cardiomyopathy	Depla [196] Paauw et al. [197] El-Ganzoury et al. [198] Topcuoglu et al. [199] Peixoto et al. [200] Patey et al. [201] Blais et al. [202] Rijpert et al. [203]	Offspring of both pregestational and gestational diabetic pregnancies had increased intraventricular septal thickness compared to controls, associated with poor maternal glycemic control. Fetuses of diabetic women exhibited significant alterations in cardiac geometry, myocardial deformation, and ventricular function that persisted following birth. Minor functional changes still found at the age of 3 years [202] disappeared among those evaluated at 7–8 years [203].
Hypoglycemia	Riskin et al. [185] Antoniou et al. [187] Yu et al. [207] Wahabi et al. [208] Feig et al. [209]	Hypoglycemia was more frequent among pregnancies with PGDM compared to GDM and both compared to controls. Maternal continuous glucose monitoring during pregnancy, but not pre-pregnancy care, resulted in fewer incidences of neonatal hypoglycemia requiring treatment with intravenous dextrose.
Polycythemia	Teramo et al. [214]	Fetal amniotic fluid erythropoietin levels were elevated in type 1 diabetic pregnancies and were independently related to low umbilical artery pH, neonatal hypoglycemia, umbilical artery pO_2_, fetal macrosomia, and growth restriction.

## Data Availability

N/A.

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
