# Peer review of "Diabetes during Pregnancy: A Maternal Disease Complicating the Course of Pregnancy with Long-Term Deleterious Effects on the Offspring. A Clinical Review"

_ijms, 2021, doi:10.3390/ijms22062965_

Round 1

Reviewer 1 Report

In this narrative review, Ornoy and colleagues extensively assess the risk of adverse obstetrical and outcomes and offspring growth/developmental abnormalities in women affected by pregestational (PGDM) and gestational (GDM) diabetes mellitus. Overall, the manuscript is nicely written, well organized, and suitable for publication. However, in view of recent research findings on this topic, I would suggest a minor revision of bibliography.

-Not only PGDM, but also GDM has been found to be significantly associated with increased risk of congenital defects of the offspring. Please revise abstract and section 2 referring to:

Diabetes Care. 2020;43(12):2983-2990. doi:10.2337/dc20-0261

J Clin Med. 2020;9(11):3553. Published 2020 Nov 4. doi:10.3390/jcm9113553

-Although usually diagnosed and treated in late-second trimester, GDM can negatively influence organogenesis and initial fetal growth, especially in women classified as at high-risk (i.e. obese or with personal history of GDM), suggesting the need of preconception counselling and anticipated OGTT testing to improve neonatal outcomes. In this respect, the initial acceleration of fetal growth related to GDM (irrespective of the woman’s risk), with predicts macrosomic births, can be already seen between 19-21 weeks of gestation, and is effectively reversed with anticipated diagnosis in high risk women and conventional timing of diagnosis (around 26 weeks) in non-high risk women. Please revise the significance of GDM in determining early-onset fetal growth abnormalities, in the appropriate sections, referring to:

Lancet Diabetes Endocrinol. 2020;8(4):292-300. doi:10.1016/S2213-8587(20)30024-3

Lancet Diabetes Endocrinol. 2020;8(7):561-562. doi:10.1016/S2213-8587(20)30189-3

J Diabetes Res. 2020;2020:5393952. Published 2020 Sep 18. doi:10.1155/2020/5393952

-To improve readability, I would suggest simplifying the columns of Table 4, in a way similar to Table 3, removing the comments’ column while expanding the description of findings.

-Given the complexity of this review’s argumentation, with many controversial study findings in each section, and consequent debates in the diabetes scientific community, I think that the conclusions section should be more detailed.

Author Response

1.We are especially thankful to this reviewer for his specific comments regarding GDM. He suggested to add several studies showing that in women with GDM there might also be an increase in congenital malformations among the offspring and that the enhanced intrauterine growth of infants of mothers with GDM may start in the second trimester of pregnancy. We indeed added these important references he mentioned in our discussion of congenital malformations (lines 118-130) and of fetal macrosomia (lines 283-312).  We also made some changes in the summary, lines 20-22.

  1. The suggestion to remove in table 4 the column of “comments” to make it similar to the other tables was well taken and the table is now similar to all others. Many of the comments were incorporated in the column describing the main findings of the studies.
  2. we expanded the conclusions, as suggested, also incorporating some comments related to GDM (lines 1111-1116).

Reviewer 2 Report

The manuscript is very long and difficult for readers to assimilate. The various sections are written in slightly different styles and there is a need for consistency. Sections 3, 4 and 5 can be better harmonised for content and style of sub-sections. A better synthesis/overhaul of the review will be necessary for readers to better appreciate this important topic.

Author Response

We thank  reviewer for his instructive opinions and comments.

This reviewer suggested to better harmonize the different sections of the manuscript. This was done, and in addition to changes in table 4 to make it similar to the other tables we also made several changes in the text. We also modified the English language whenever needed.

Reviewer 3 Report

This is a well written, interesting article that will contribute to the literature.

Author Response

We thank this reviewer for his positive and warm comments.